# Extraction of Nanocellulose from the Residue of Sugarcane Bagasse Fiber for Anti-*Staphylococcus aureus* (*S. aureus*) Application

**DOI:** 10.3390/polym16111612

**Published:** 2024-06-06

**Authors:** Krairop Charoensopa, Kotchaporn Thangunpai, Peifu Kong, Toshiharu Enomae, Wat Ploysri

**Affiliations:** 1Department of Industrial Arts and Science, Faculty of Engineering and Industrial Technology, Suan Sunandha Rajabhat University, 1 U Thong Nok Rd, Dusit, Bangkok 10300, Thailand; krairop.ch@ssru.ac.th; 2Graduate School of Science and Technology, University of Tsukuba, 1-1-1 Tennodai, Tsukuba 305-8572, Ibaraki, Japan; s2036028@u.tsukuba.ac.jp (K.T.); peifukong@163.com (P.K.); 3Institute of Life and Environmental Sciences, University of Tsukuba, 1-1-1 Tennodai, Tsukuba 305-8572, Ibaraki, Japan

**Keywords:** nanocellulose, sugarcane bagasse, characterization, anti-*Staphylococcus aureus* analysis

## Abstract

Nanocellulose contains a large number of hydroxyl groups that can be used to modify its surface due to its structure. Owing to its appealing features, such as high strength, great stiffness, and high surface area, nanocellulose is currently gaining popularity in research and industry. The extraction of nanocellulose from the leftover bagasse fiber from sugarcane production by alkaline and acid treatment was successful in this study, with a production yield of 55.6%. The FTIR and XPS results demonstrated a difference in the functional and chemical composition of untreated sugarcane bagasse and extracted nanocellulose. SEM imaging was used to examined the size of the nanocellulose with ImageJ software v1.8.0. TGA, DTG, and XRD analyses were also performed to demonstrate the successful extraction of nanocellulose in terms of its morphology, thermal stability, and crystal structure before and after extraction. The anti-*S. aureus* activity of the extracted nanocellulose was discovered by using an OD_600_ test and a colony counting method, and an inhibitory rate of 53.12% was achieved. According to the results, nanocellulose produced from residual sugarcane bagasse could be employed as an antibacterial agent.

## 1. Introduction

Approx. 80% of the world’s sugar comes from sugarcane, which is grown in 120 countries, producing an average annual yield of 1.8–2.0 billion tons. In 2019, Thailand was the fourth-largest producer of sugar, contributing 8.10% to the global overall sugar production. In addition, it was the second-largest exporter of sugar, representing 16.95% of the world’s sugar exports and reaching a value of USD 2.97 billion. Sugarcane production is becoming increasingly important in Thai agriculture at the national level [1]. After crushing sugarcane stalks to obtain juice for sugar manufacture, 32.4% of the bagasse and molasses become waste every year [2], also producing a plant-based byproduct called lignocellulosic biomass. This is a beneficial byproduct with a variety of applications, including dairy feed [3], biofuel [4], bioethanol [5], renewable energy [6], paper and pulp manufacture [7], and so on [8]. Cellulose, lignin, and hemicellulose are all significant components of lignocellulosic biomass. These components can be recovered from residue using a variety of conventional and modern processes and have applications in a variety of food processing businesses. The proportions of cellulose, hemicellulose, and lignin in lignocellulosic biomass vary based on the source of waste. Typically, agricultural waste contains 35–50% cellulose, 25–50% hemicellulose, and 5–15% lignin [9,10].

Cellulose is one the most abundant biopolymers on Earth and serves as the primary reinforcing material in plant structures. For more than 150 years, materials based on cellulose and its derivatives have been employed in a wide range of applications, including food, paper manufacture, biomaterials, and pharmaceuticals. Furthermore, natural cellulose-based materials, such as wood, hemp, cotton, and linen, have been applied as engineering materials in society for thousands of years. A large number of industries throughout the world engaged in the production of forest-based products, paper, textiles, and so on attest to the material’s ongoing popularity [11,12,13]. Cellulose extraction can be accomplished by applying three methods: mechanical, chemical, and bacterial. Mechanical methods for extracting cellulose include high-pressure homogenization, milling, pulverizing, and steam explosion. Chemical retting, acid retting, alkali treatment, and degumming are chemical extraction procedures. For bacterial cellulose, a range of techniques are employed, for example, static, agitated, or shaking procedures, bioreactor-based cultivation approaches, etc. Depending on the intended use, there is a variety of sizes for cellulose extraction. In fact, microcellulose and nanocellulose are commonly utilized sizes in industrial applications [14].

Nanocellulose is described as a cellulose nanomaterial that has one or more nanoscale dimensions. The morphology and qualities of nanocellulose depend on its natural source, different pretreatments, and the separation and extraction processes. There are several methods to extract nanocellulose from natural fibers, for example, chemo-mechanical, cryocrushing, acid hydrolysis, etc. [15]. Recently, nanocellulose generated from sugarcane bagasse has emerged as one of the most widely used sources raw materials because a large proportion of cellulose (40–50%) contained in sugarcane bagasse is a valuable resource. It is currently gaining popularity in research and industry due to its attractive properties, such as its high strength, excellent stiffness, and high surface area [16]. Owing to its structure, nanocellulose contains a large number of hydroxyl groups that are available for surface modification. Nanocellulose has numerous applications in our daily lives, including biomedical devices [17], nanocomposite materials [18], textiles [19], antibacterial [20], and so on [21]. Nanocellulose for food packaging is currently reported as one of the advantages of its use. Since the most significant function of food packaging is ensuring the quality and safety of food throughout storage and food logistics, it is crucial to prevent spoilage caused by microorganisms, chemical contaminants, water vapor, oxygen, carbon dioxide, volatile compounds, moisture, and exposure to light and physical forces in order to extend the shelf life of food products. Accordingly, packaging materials serve as physical protection and generate appropriate physicochemical conditions to ensure food quality throughout storage [22].

In both developing and developed nations, food poisoning has been recorded with diverse causal factors. *Staphylococcus aureus* (*S. aureus*) is a significant pathogen of food poisoning that can arise in both sporadic and epidemic forms. It has the potential to cause life-threatening infections in children, the elderly, and immunocompromised people [23,24]. Therefore, researchers are interested in improving the avoidance of *S. aureus* transmission in food. Antibacterial food packaging has emerged as an effective and promising packaging technology. Several active substances have been effectively included in antibacterial packaging, including silver, gold, and zinc oxide nanoparticles, etc. [25,26,27]. In addition, several antimicrobial polymers are used in antibacterial applications, such as chitosan and its derivatives, which are effective antibacterial agents. Antimicrobial materials based on nanocellulose can be employed in a variety of applications, including drug carriers, packaging materials, and wound care products [28].

In this study, we conducted modifications and extractions of nanocellulose from residual sugarcane bagasse fibers sourced from Northeastern Thailand, which is renowned as the prime cultivation region for *Saccharum officinarum*. These efforts were aimed at formulating innovative anti-*S. aureus* compounds suitable for integration into food packaging materials.

## 2. Materials and Methods

### 2.1. Materials

Residual sugarcane bagasse fiber was sourced from Thong Pha Phum Fruit and Vegetable Community Enterprise located in Kanchanaburi, Thailand. Sodium hydroxide (NaOH) and hydrochloric acid (HCl) were purchased from S.N.P. General Trading Co., Ltd., (Bangkok, Thailand) and utilized for this experiment. Ethanol (99.5%, guaranteed reagent) was provided by Fujifilm Wako Chemicals (Osaka, Japan), while supplementary solvents and chemicals were analytical grade and employed without additional purification steps. Microbiologics, Inc. (Saint Cloud, MN, USA) provided *S. aureus* derived from ATCC 25923. Geno Technology Inc. (Saint Louis, MO, USA) provided the tryptic soy agar (TSA) and tryptic soy broth (TSB). Greiner Bio-One International GmbH (Kremsmünster, Austria) provided 96 mm-diameter Petri dishes for culture.

### 2.2. Extraction of Nanocellulose from Residual Sugarcane Bagasse Fiber

To gain the nanocellulose from the residual sugarcane bagasse fiber, the extraction method from Gond et al. [29] was used with modifications. In the sample preparation step, the residual sugarcane bagasse fiber was prepared and washed with DW water to remove contamination from the post-harvest process. Subsequently, all the fibers underwent drying in an oven for 2 or 3 days at 80 °C to eliminate any moisture adsorbed on the fibers. Initially, 10 g of the dried fibers were immersed in a 15%wt NaOH solution at room temperature for 4 h. Following the 4 h immersion period, the fibers were washed repeatedly with DW water to remove any remaining alkali from the fibers. This step aimed to eliminate the presence of lignin, wax, and natural oils that covered the fibers’ surface. Additionally, this process increased the surface roughness of the fibers. Afterwards, the fibers were oven-dried at 80 °C for 1 d. In the second step, the fibers treated with alkali were immersed in a 1 M HCl solution to help break up the cell wall and separate the microfibrils at 80 °C for 4 h. Following this procedure, the HCl fibers were thoroughly rinsed with DW to get rid of any remaining residue HCl, and then dried in the oven for 1 d at 80 °C. Next, 2 wt% NaOH solution was added to the acid-treated fibers for 4 h at 80 °C to remove any residual non-cellulosic materials from the fibers. After that, the fibers underwent another round of washing and subsequent drying in the oven. Finally, the nanocellulose was obtained by subjecting the treated fibers to a high-speed grinder for 3 min.

### 2.3. Yield Percentage

The percentage yield of the extracted nanocellulose was calculated through the gravimetric method, as shown in Equation (1) [30]:(1)$$\mathrm{The}\;\mathrm{yield}\;\mathrm{percentage}\;\mathrm{of}\;\mathrm{the}\;\mathrm{extracted}\;\mathrm{nanocellulose}=\frac{m_{2}}{m_{1}}\times 100$$
where *m*_1_ represents the initial weight of the sugarcane bagasse fiber, and *m*_2_ is the final weight of the nanocellulose obtained.

### 2.4. Characterization

#### 2.4.1. Fourier Transform Infrared Spectroscopy (FTIR)

The functional properties of the untreated bagasse fiber and nanocellulose were measured using Fourier transform infrared spectroscopy (FTIR-6100, JASCO Corporation, Tokyo, Japan) combined with ATR (PRO ONE PKS-Z1, JASCO Corporation, Tokyo, Japan) spectrometry using a Ge prism (PKS-G1, ASCO Corporation, Tokyo, Japan) in the range of 500–4000 cm^−1^.

#### 2.4.2. Field-Emission Scanning Electron Microscopy (FESEM)

All the samples underwent platinum (Pt) coating using a spatter prior to examination with FESEM (SU8020, Hitachi, Tokyo, Japan).

#### 2.4.3. Thermal Analysis

Thermogravimetric analysis (TGA) was performed utilizing a thermogravimetric analyzer (TG/DTA7300, Seiko Instruments Inc., Chiba, Japan). The temperature was raised to 550 °C with a heating rate of 15 °C/min, while argon gas flowed at a rate of 200 mL/min.

#### 2.4.4. X-ray Diffraction (XRD)

The XRD profiles of each sample, including raw bagasse fiber and nanocellulose, were analyzed using XRD (D8 Advanced, AXS-Bruker, Karlsruhe, Germany). The measurements were conducted over a 2θ range from 10° to 60° at 40 kV and 40 mA, employing a Cu-Kα radiation (1.5418 Å) source. The inter-planar separation (d-spacing) between atoms was calculated using Bragg’s Law, expressed as Equation (2) [31]:(2)nλ=2dsinθ
where *n* (an integer) is the order of reflection, *λ* is the wavelength of the incident X-rays (0.154 nm), *d* is the d-spacing, and *θ* is the angle of incidence.

The crystallinity index (C_I_) was determined using the peak height approach, calculated according to Equation (3) [32]:(3)$$C_{I}(\%)=\left(\frac{I_{C}-I_{am}}{I_{C}}\right)\times 100$$
where, *I_C_* and *I_am_* represent the intensity of the crystalline peak (002) and the intensity attributed to the amorphous peak, respectively.

#### 2.4.5. X-ray Photoelectron Spectroscopy (XPS)

The surface compositions of the specimens underwent additional analysis through XPS (JPS-9010TR, JEOL, Tokyo, Japan) with an AlKα radiation (1486.6 eV) source at 10 kV and 20 mA. The high-resolution peaks of C_1s_ and O_1s_ were evaluated using XPSpeak41 v4.01. To avoid the charging effect, the C_1s_ peaks of all samples were calibrated by shifting to 248.8 eV. A Shirley-type background function was employed for spectrum fitting [33].

### 2.5. Anti-S. aureus Analysis

The extracted nanocellulose demonstrated anti-*S. aureus* activity, and the activity was evaluated using an OD_600_ test and a colony counting method, as described in our previous study [34]. The experimental details are as follows: Firstly, a glass tube (named the sample tube) containing 4 mL TSB and 0.08 g of extracted nanocellulose was sterilized using an autoclave (LBS-245, TOMY, Tokyo, Japan). For comparison, a control glass tube (named the control tube) only containing 4 mL TSB was prepared. Subsequently, a pre-diluted *S. aureus* inoculum was added to the sample and control tubes to achieve a final *S. aureus* concentration of 1 × 10^4^ cells mL^−1^. Afterwards, the sample and control tubes were placed in a bio-shaker (BR-23FH, Taitec Corporation, Tokyo, Japan) for the cultivation of *S. aureus* at a speed of 100 rpm min^−1^ and a temperature of 30 °C for 20 h. Finally, the OD_600_ values of the cultivated *S. aureus* inoculums in the sample and control tubes were measured using an ultraviolet-visible spectrophotometer (U-3900, HITACHI, Tokyo, Japan). Additionally, the cultivated *S. aureus* inoculums in both the sample and control tubes were diluted 10^6^ times, respectively. Then, 20 μL of these diluted inoculums were spread onto TSA plates (sample and control plates), respectively, followed by incubation at 37 °C for 24 h. The number of *S. aureus* colonies on the TSA plate was then calculated. The inhibitory rate was determined using the following formula:$$\mathrm{Inhibitory}\;\mathrm{rate}\;(\%)=\frac{C-S\;}{C}\times \;100$$
where S and C are the numbers of *S. aureus* colonies in the sample and control plates, respectively.

### 2.6. Statistical Analysis

All values in this study were conducted in triplicate and reported as the means ± standard deviation. The data were subjected to a one-way analysis of variance (ANOVA) in SPSS v20. Statistical significance was at *p* < 0.05 using the least significant difference (LSD) test.

## 3. Results and Discussion

### 3.1. Yield Percentage of Nanocellulose

The yield percentage was calculated using Equation (1), where an initial weight of 10 g of the bagasse fiber was measured and a final weight of 5.59 g of the nanocellulose was obtained after all the extraction steps. The yield percentage of nanocellulose was 55.9%. To eliminate the hemicellulose and lignin from sugarcane bagasse fiber, the alkali and acid treatment were efficiently removed [35]. The percentage yield was lower due to the presence of 30–70% hemicellulose and lignin in the sugarcane bagasse produced in Thailand [36]. The alkaline and acid treatment effectively removed the lignin, wax, and natural oils, as well as the residue of non-cellulosic components from the sugarcane bagasse. Lignin and hemicellulose were additionally removed during the second alkaline bleaching process [29]. The lower yield percentage of nanocellulose extraction was supported by all of the explanations.

### 3.2. FTIR

The changes in the functional properties of both the bagasse fiber and nanocellulose were identified using FTIR, as shown in Figure 1. Bagasse fiber consists of three main components: cellulose, hemicellulose, and lignin. These constituents are composed of alkanes, ketones, esters, alcohols, and aromatics with distinct oxygen-containing groups [37]. The broad peak at 2996–3703 cm^−1^ indicates the OH stretching vibration of the hydroxyl group in the cellulose, hemicellulose, and lignin [38]. Additionally, the C-H group, primarily found in the structures of cellulose, hemicellulose, and lignin, can be observed from 2776 to 2998 cm^−1^. In addition, a peak at 1732 cm^−1^ represents the C=O structure of the untreated bagasse fiber, indicating the C=O stretching vibration of the acetyl and uronic ester groups present in the hemicellulose and lignin structure. However, after the extraction of nanocellulose, the peak at 1732 cm^−1^ disappeared, confirming the successful removal of hemicellulose and lignin [39]. Additionally, the aromatic skeletal vibration of lignin in the untreated bagasse fiber was found at 1250, 1375, and 1518 cm^−1^ [40]. After the extraction of nanocellulose, the intensity of the peak at 1250 cm^−1^ was significantly decreased due to the removal of lignin [41]. In addition, the peak at the 1033 cm^−1^ increased after the extraction of nanocellulose, signifying the improvement of the surface area of the nanocellulose. The OH bending vibration of adsorbed moisture appeared at 1637 cm^−1^ due to the hydrophilic properties of the fiber. Nevertheless, this peak was not detected in the nanocellulose after extraction [42]. Furthermore, the broad peak of the OH stretching vibration of nanocellulose at 3332 cm^−1^ was found to decline after the extraction. This supports the decrease in the adsorption of moisture [43].

### 3.3. FESEM

The results of the morphological analysis using FESEM are shown in Figure 2a, representing the cellulose of the bagasse fiber; Figure 2b illustrates the nanocellulose obtained from the extraction process from the bagasse fiber. Figure 2a displays the bagasse fiber before the extraction process. It reveals that the cellulose of the bagasse fiber was surrounded by the remaining materials and firmly stuck together in a pile. This observation may be attributed to the presence of a wax layer, intact lignocellulose fragments, and a large number of microfibrils. The nanocellulose treated with alkaline and acid hydrolysis after the extraction process is shown in Figure 2b. The surface morphology of the nanocellulose was a higher quality compared to the cellulose products. This superior quality can be attributed to the treatment steps involved, including alkaline dewaxing, acid hydrolysis, and the second alkaline bleaching process, which led to the formation of microfibrils by removing the lignin and hemicellulose. The nanocellulose in Figure 2b was measured using ImageJ software v1.8.0 to determine its length, diameter, and aspect ratio (L/D) of ten random areas. The mean measurements of the length, diameter, and L/D were 442.9, 142.2, and 3.12 nm., respectively (Table 1). The given nanocellulose has a favorable aspect ratio, which indicates a higher potential for reinforcing. Hence, the morphological analysis provides evidence that nanoparticles can be separated from bagasse using the method presented in this study [29].

### 3.4. XRD

The XRD patterns of both the cellulose from the bagasse fiber and the nanocellulose are depicted in Figure 3. Cellulose, hemicellulose, and lignin are the primary components of bagasse fiber. The crystalline structure of cellulose arises from its hydrogen bond structure. Hemicellulose and lignin, on the other hand, have an amorphous structure [44]. Two remarkable peaks of cellulose from bagasse fibers were exhibited at 16.3° and 22.1°, corresponding to the (1–01) and (002) planes, respectively. These peaks are characteristic of cellulose type 1, commonly observed in various types of fiber. Following the extraction of nanocellulose, a slight shoulder was clearly identified at 15.6°, corresponding to the (110) plane, showing success in removing the amorphous structure of cellulose from the bagasse pulp [29,45]. Furthermore, the comparison of the crystallinity index (%C_I_) between cellulose and nanocellulose from the bagasse fiber extraction process was calculated and confirmed using Equation (3). Cellulose and nanocellulose had %C_I_ of 31.26 and 54.96, respectively. The %C_I_ of nanocellulose after the alkaline and acid treatments was higher than that of the untreated bagasse fiber. This was a result of the breakdown of the amorphous area and the enhancement of the crystalline regions. Furthermore, the alkali treatment improved the C_I_ by excluding amorphous non-cellulosic components and forming H-bonding, which restricts the free mobility of cellulose chains. The removal of non-cellulosic elements influenced the crystallinity properties [42,46].

### 3.5. Thermal Analysis

The percentage change in weight with temperature for both cellulose and nanocellulose from the bagasse fiber extraction process was determined with the thermal analysis (TGA and DTG), as depicted in Figure 4a,b. The decomposition of bagasse fiber occurs at different temperatures ranges: moisture evaporation (50–100 °C), cellulose (229–344 °C), hemicellulose (170–229 °C), and lignin (229–462 °C) [47]. The weight loss percentage was noted at two primary degradation stages: the first, occurring at around 50–100 °C was associated with moisture loss (2%), while the second, representing 96.3% of the weight loss, occurred between 170 and 344 °C. This latter stage involved at least two reactions (derivative curve), corresponding to the thermal decomposition of cellulose and hemicellulose, which are the major components of bagasse fiber. Furthermore, the lignin peak is broader and appears between 229 °C and 462 °C overlapping with the peaks of cellulose and hemicellulose [48]. Figure 4a compares the TGA of cellulose and nanocellulose, with the initial temperature stage ranging from 100 to 229 °C (6.5%) due to the evaporation of free moisture from the surface of the nanocellulose. This phenomenon is caused by the elimination of hemicellulose and lignin post-extraction. Additionally, the nanocellulose also shows the highest thermal stability [49]. At temperatures exceeding 229 °C, the second stage shows a decrease in weight (93.5%) due to the decomposition of the fiber components, such as cellulose, hemicellulose, and lignin. The results indicate that the bagasse fiber, after the nanocellulose extraction process, had lower thermal stability compared to the untreated fiber. This reduction in thermal stability is attributed to the removal of the hemicellulose and lignin, which serve as protective thermal barriers. However, the results also indicate that the weight loss of the nanocellulose after extraction was lower than that of the cellulose before extraction [50].

### 3.6. XPS

The chemical compositions of the cellulose and nanocellulose from the bagasse fiber extraction process were determined using XPS. The full XPS spectra of C_1s_ and O_1s_ were compared before and after the extraction of the nanocellulose, confirming the FTIR results. Figure 5 shows the full XPS spectra of both of the cellulose and nanocellulose from the bagasse fiber, with two associated peaks attributed to C_1s_ (approximately 287 eV) and O_1s_ (approximately 534 eV).

Figure 6a–d depicts the C_1s_ and O_1s_ spectra of the cellulose and nanocellulose from the bagasse fiber. The C_1s_ spectrum of the cellulose from bagasse fiber (Figure 6a) corresponds to three groups: -CHx(287.1 eV), -C-O- (289.1 eV), and -C(=O)-O- (290.4 eV). For the nanocellulose (Figure 6c), an additional of C_1s_ peak was detected C-C=O (290.7 eV), which confirms the successful extraction of nanocellulose [51]. Furthermore, three O_1s_ peaks of the cellulose from the bagasse fiber (Figure 6b) appear at 535.3, 534.5 and 533.4 eV, corresponding to -O-, -OH and -C=O-, respectively. However, in the nanocellulose obtained after extraction, the O_1s_ peaks show a slight shift, and only -O- (535.2 eV) and -OH (533.8 eV) (Figure 6d). This is consistent with previous research, confirming the successful extraction of the nanocellulose [52].

### 3.7. Anti-S. aureus Activity

The anti-*S. aureus* analysis of the nanocellulose was carried out via an OD_600_ test and a colony counting method. The results of the OD_600_ values are presented in Table 2. Clearly, the OD_600_ value of *S. aureus* inoculum in the medium containing nanocellulose was lower than that of the control. This inhibitory effect the growth of *S. aureus* was likely due to the antiadhesion effect of the nanocellulose, as adhesion is the initial stage of bacterial biofilm formation [53,54]. The bacterial surfaces, which typically have a negative charge, can interact with extracellular carriers through several mechanisms, such as van der Waals forces, steric effects, and electrostatic interactions. Nevertheless, the carboxyl group presenting in nanocellulose can dissociate and form a negatively charged carboxylate anion. This anion can induce electrostatic repulsion toward bacterial cells, thus impeding their attachment to surfaces and disrupting the formation of bacterial biofilms [55]. Furthermore, nanocellulose with a lower molecular weight and an extremely small size has an enhanced diffusion property and better interaction with bacteria, thus likely leading to an improved effect of bacteriostasis. As a result, the death of some *S. aureus* cells occurred in the medium containing nanocellulose during incubation, resulting in a slight increase in the OD_600_ value. Therefore, to obtain a more precise evaluation, a colony counting method was conducted, determining an inhibitory rate of 53.12 ± 7.13%. This outcome suggests that nanocellulose extracted from bagasse fiber holds potential for anti-*S. aureus* applications.

## 4. Conclusions

This study successfully developed nanocellulose extracted from residual sugarcane bagasse through alkaline and acid treatments, achieving a production yield of 55.6%. The FTIR and XPS analyses indicated differences in both the functional and chemical compositions between the untreated sugarcane bagasse and the nanocellulose after the extraction. The particle size of the nanocellulose was affirmed by FE-SEM analysis and calculated using ImageJ software v1.8.0. Additionally, TGA, DTG, and XRD analyses were also used and confirmed the successful extraction of nanocellulose by assessing its morphological characteristics, thermal stability, and crystal structure before and after extraction. Significantly, an inhibitory rate of 53.12% against *S*. *aureus* was achieved, suggesting that nanocellulose derived from sugarcane bagasse has potential as an antibacterial agent.

## Figures and Tables

**Figure 1 polymers-16-01612-f001:**
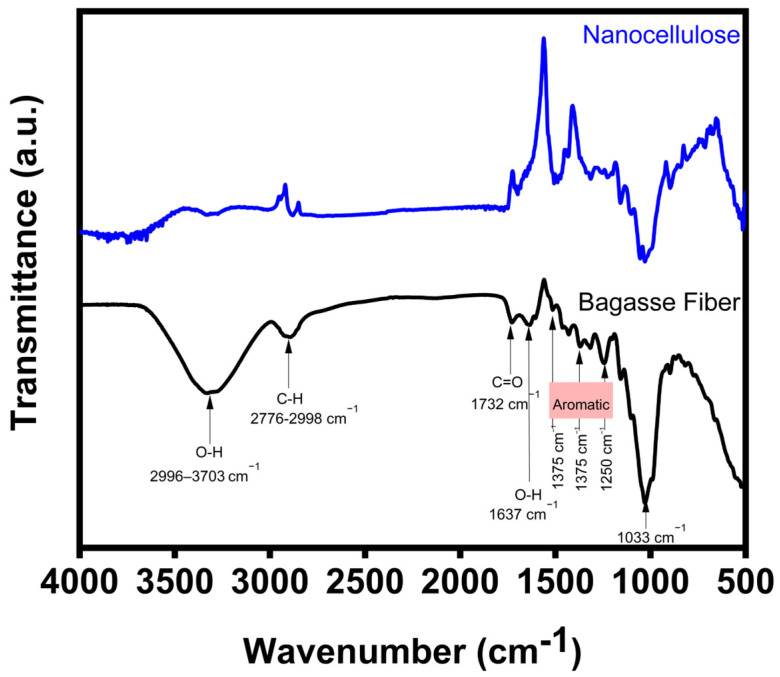
FTIR spectra of bagasse fiber compared to the extraction of nanocellulose.

**Figure 2 polymers-16-01612-f002:**
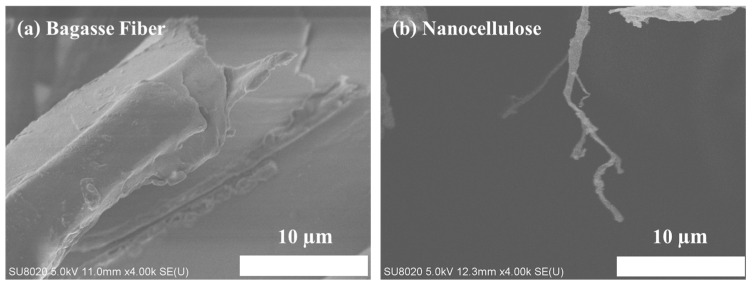
FESEM images of (**a**) cellulose of the bagasse fiber and (**b**) nanocellulose after the extraction.

**Figure 3 polymers-16-01612-f003:**
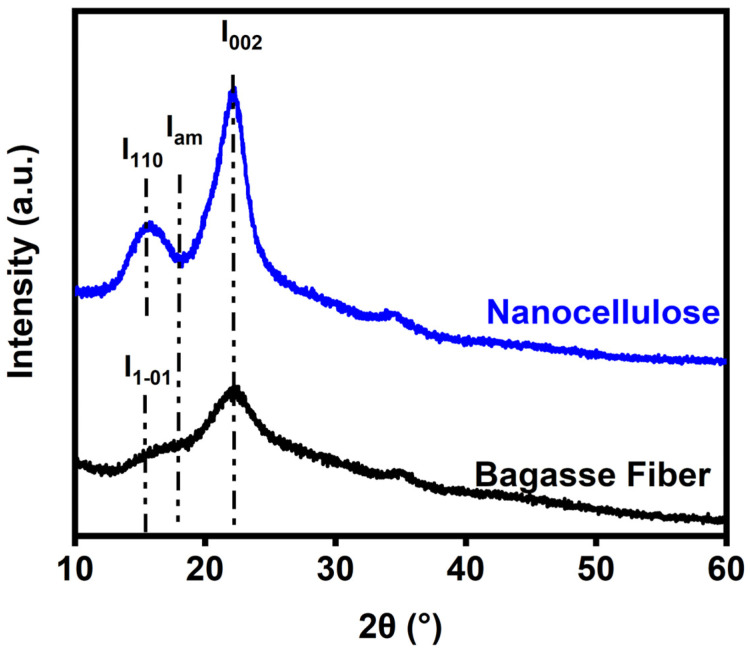
XRD of cellulose of the bagasse fiber and nanocellulose after the extraction.

**Figure 4 polymers-16-01612-f004:**
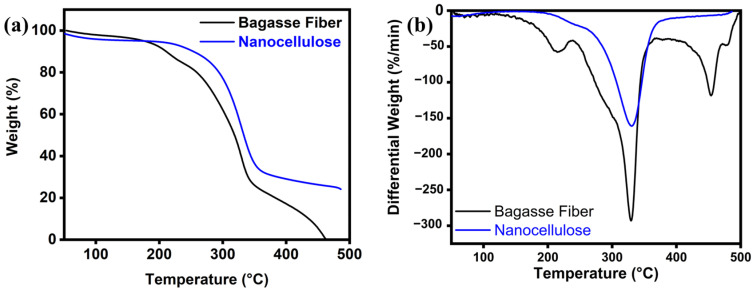
(**a**) TGA and (**b**) DTG of cellulose of the bagasse fiber and nanocellulose after the extraction.

**Figure 5 polymers-16-01612-f005:**
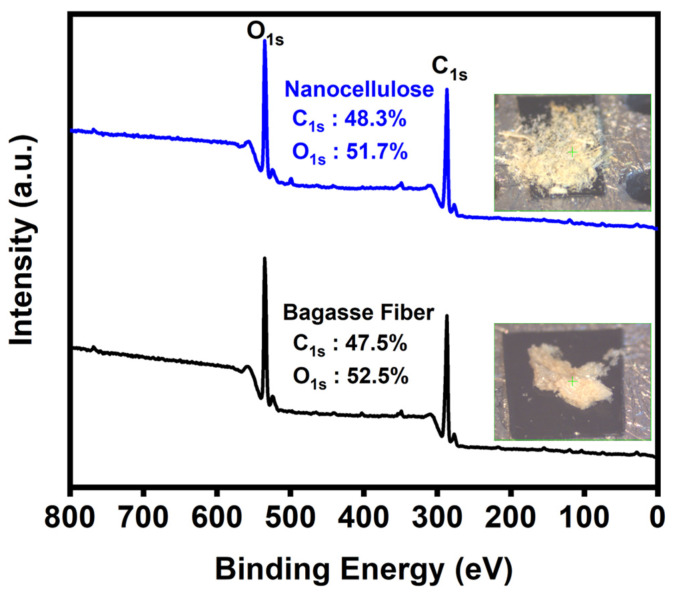
XPS full spectra of cellulose of the bagasse fiber and nanocellulose after the extraction.

**Figure 6 polymers-16-01612-f006:**
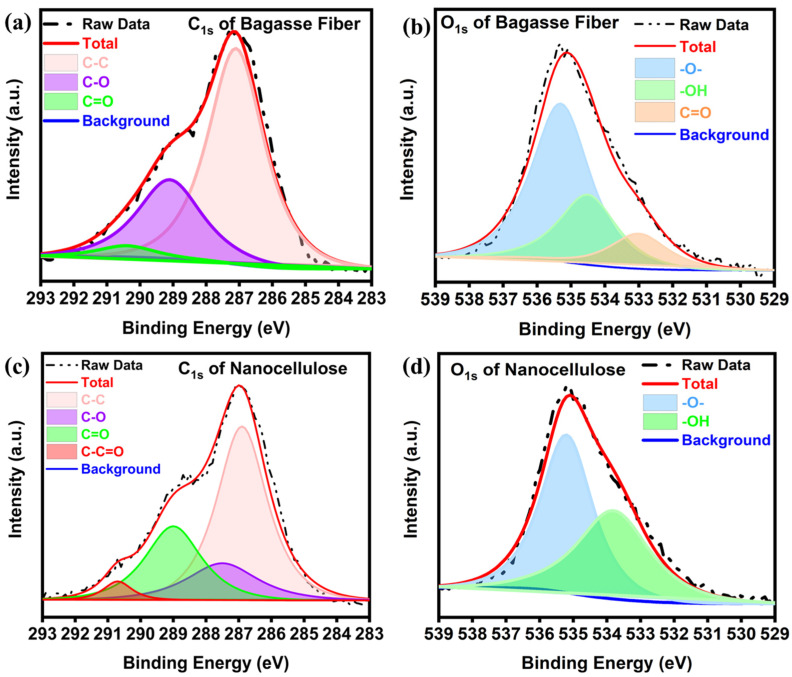
High-resolution XPS spectra of the (**a**) C_1s_ and (**b**) O_1s_ of bagasse fiber, and the (**c**) C_1s_ and (**d**) O_1s_ of nanocellulose after the extraction.

**Table 1 polymers-16-01612-t001:** Particle sizes and aspect ratios of nanocellulose extracted from sugarcane bagasse.

No.	Length (nm)	Diameter (nm)	Aspect Ratio (L/D)
1	711	43	16.53
2	230	27	8.52
3	281	172	1.63
4	129	12	10.75
5	500	160	3.13
6	352	203	1.73
7	906	742	1.22
8	887	47	18.87
9	285	12	23.75
10	148	4	37.00
Average	442.9	142.2	3.11

**Table 2 polymers-16-01612-t002:** OD_600_ values of *S. aureus* inoculum in the medium with and without nanocellulose.

	With Nanocellulose	Without Nanocellulose (Control)
OD_600_ value	2.10 ± 0.06	3.44 ± 0.14

Note: OD600 of 1.0 ≈ 8 × 10^8^ cells mL^−1^. Significant difference of two values is shown according to the LSD test (*p* < 0.05).

## Data Availability

Data are contained within the article.

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
