# Peer review of "Extraction of Nanocellulose from the Residue of Sugarcane Bagasse Fiber for Anti-Staphylococcus aureus (S. aureus) Application"

_polymers, 2024, doi:10.3390/polym16111612_

Round 1
Reviewer 1 Report
Comments and Suggestions for Authors
The manuscript entitled, "Extraction nanocellulose from Residue Sugarcane bagasse Fiber for anti-Staphylococcus aureus (S. aureus) application" lacks in comprehensive experimental design and novelty. Language expression is very poor, please refer to specific comments below:
Abstract: The abstract in its current form is not acceptable, atleast write what methods were used and what were results in terms of numerical terms. Terms like notable, successful, slow growth such terms are very vague without numerical terms to support the findings. Moreover, the writing expression is not good. The low number of colonies correspond to absorbance at 600 nm?? how??
Introduction: Background information is not sufficient and lack in depth support from recent literature.
Section 2.2, nanocellulose was modified and extracted??? how come?
what are the components other than nanocellulose, authors should have quantified.
Add a section on statistical analysis.
How authors were sure that nanocellulose was produced rather than cellulose?? why not measure the mean particle diameter?? From SEM it does not look like nano size.
The all three replicates in table 1 do not make any sense. Moreover, from plates it does not show any antibacterial potential. How authors made predictions.
As mentioned antibacterial potential as a major claim of the study in conclusion but results do not justify what authors are reporting here.
Comments on the Quality of English LanguageNeeds rewriting of some parts.
Reviewer 2 Report
Comments and Suggestions for Authors Converting of agricultural wastes to value-added products is an emerging trend of modern science and technology. One of the most abundant byproducts on a worldwide scale is sugarcane waste. Despite of quite active recycling of this product, there is still wide space to find additional applications. The present work exploring the ways to extract nanocellulose from untreated sugarcane bagasse as a candidate for antibacterial treatment partly fills this gap and makes a certain input into the field. The authors suggest some chemical ways to extract nanocellulose that appear to be quite efficient. The latter conclusion is based on thorough characterization of the composition, structure, morphology and the antibacterial effect of nanocellulose via wide set of complimentary experimental tools. The main outcome of the work, that the nanocellulose produced from residue sugarcane bagasse can be employed as an antibacterial agent, is of high applied relevance, beginning from the food packaging and to the wound treatment and recovery. The work is relevant for publication, but before that there are some points to address: 1) It is difficult to compare description part of the FTIR spectra (section 3.2) with the Fig. 1. It is not ambiguous to follow specific wavenumbers, outlined in the text, with the spectral features. The Fig. 1 should be redrawn in the form indicating the value of the wavenumber by the specific vibrational band. Also, the fact that there are peaks in Fig. 1 directed to opposite directions (up and down, transmittance, absorbance) needs clarification. 2) The fine component analysis of XPS O1s and C1s peaks (section 3.6) needs clarification in terms of the of possible influence of the charging effect. The studied sample is likely insulating, so the charging effect may change the position and the fine structure of the photoelectron line. Since the authors' conclusions rely heavily on this XPS part, clarification is necessary. 3) Although the text is understandable, some grammar refinement throughout the entire text is required. Comments on the Quality of English Language Although the text is understandable, some grammar refinement throughout the entire text is required.Author Response
Please see the attachment

Reviewer 3 Report
Comments and Suggestions for Authors
The manuscript needs major amendments before being considered for publication in Polymers.
1. The title is without preposition and conjunctions. Correct the title as "Extraction of nanocellulose from the residue of sugarcane bagasse fiber for anti-Staphylococcus aureus (S. aureus) application."
2. The grammar of the manuscript must be improved significantly.
3. 29% similarity text found, which is not acceptable.
4. The main findings, i.e., anti-aureus activity, lack significant details. Show us disc diffusion, percentage inhibition, MIC, MBC, and other antibacterial tests that will help us be sure that nanocellulose has potential.
Comments on the Quality of English LanguageMust be improved
Round 2
Reviewer 3 Report
Comments and Suggestions for Authors
The authors have revised the manuscript as per the reviewers suggestions. Hence I recommend this paper for publication.